# Meiotic sex chromosome cohesion and autosomal synapsis are supported by *Esco2*

François McNicoll[1,*], Anne Kühnel[1,*], Uddipta Biswas[1], Kai Hempel[1], Gabriela Whelan[2], Gregor Eichele[2], Rolf Jessberger[1]

In mitotic cells, establishment of sister chromatid cohesion requires acetylation of the cohesin subunit SMC3 (acSMC3) by ESCO1 and/or ESCO2. Meiotic cohesin plays additional but poorly understood roles in the formation of chromosome axial elements (AEs) and synaptonemal complexes. Here, we show that levels of ESCO2, acSMC3, and the pro-cohesion factor sororin increase on meiotic chromosomes as homologs synapse. These proteins are less abundant on the largely unsynapsed sex chromosomes, whose sister chromatid cohesion appears weaker throughout the meiotic prophase. Using three distinct conditional *Esco2* knockout mouse strains, we demonstrate that ESCO2 is essential for male gametogenesis. Partial depletion of ESCO2 in prophase I spermatocytes delays chromosome synapsis and further weakens cohesion along sex chromosomes, which show extensive separation of AEs into single chromatids. Unsynapsed regions of autosomes are associated with the sex chromatin and also display split AEs. This study provides the first evidence for a specific role of ESCO2 in mammalian meiosis, identifies a particular ESCO2 dependence of sex chromosome cohesion and suggests support of autosomal synapsis by acSMC3-stabilized cohesion.

## Introduction

In most organisms, the generation of haploid gametes requires one genome replication followed by two cell divisions. In diploid organisms, the four initial copies of each chromosome, that is two pairs of sister chromatids, are thereby reduced to one chromatid per cell by the process of meiosis. Meiosis features unique chromosome structures and behaviour, most obvious during the first meiotic division. Prophase I starts with leptonema, where meiosis-specific proteins such as SYCP3 assemble on each pair of sister chromatids to initiate the formation of axial elements (AEs), compact chromosome cores from which chromatin loops emerge. AEs are completed and the two homologous AEs of each chromosome start to pair in zygonema and

form the synaptonemal complex (SC), harboring four sister chromatids. This process depends on DNA double-strand break (DSB)–induced homologous recombination. Formation of the SC between paired homologous chromosomes, a process referred to as synapsis, is complete in pachynema except for the two sex chromosomes in males. Their X and Y chromosomes are heterologous and in mice pair only through a short, ~700-Mbp-long stretch at the centromere-distal end called the pseudoautosomal region (PAR) (Solari, 1970). Repair of DSBs on the X and Y chromosomes is slower than on autosomes, and the unsynapsed sex chromosomes are transcriptionally silenced. Unsynapsed prophase I chromosomes carry chromatin marks such as phosphorylated histone H2AX ($\gamma$H2AX), SUMO-1, and HORMAD-1, which disappear from autosomes upon synapsis but stay on X/Y chromosomes as they remain unsynapsed along most of their length (for reviews on aspects of meiotic chromosome structure and dynamics see Burgoyne et al (2009), Handel and Schimenti (2010), Zickler and Kleckner (2015), Bolcun-Filas and Handel (2018), Gao and Colaiacovo (2018), Link and Jantsch (2019)).

In meiosis, to ensure sister chromatid cohesion, to generate a proper axial-loop chromosome structure, and to allow synapsis and associated processes, the cohesin complex is required. In mitotic cells, this ring-shaped complex is composed of SMC1$\alpha$, SMC3, RAD21, and one variant of SCC3 (either STAG1/SA1 or STAG2/SA2). In addition to these canonical subunits, meiocytes express several meiosis-specific components: one variant of SMC1$\alpha$ (SMC1$\beta$), two variants of RAD21 (RAD21L and REC8), and an additional variant of SCC3 (STAG3/SA3) (reviewed in Nasmyth and Haering (2009), Wood et al (2010), Nasmyth (2011), Haering and Jessberger (2012), Seitan and Merkenschlager (2012), McNicoll et al (2013), Remeseiro and Losada (2013), Rankin (2015), Lee (2017), Ishiguro (2019)).

Pro- and anti-cohesion factors determine the persistence time of cohesin on chromosomes and the ability of cohesin to confer sister chromatid cohesion as opposed to other functions such as regulation of gene expression. Acetylation of SMC3 (acSMC3) is required for cohesion establishment, and mammalian cells express two SMC3 acetyltransferases, ESCO1 and ESCO2 (Rolef Ben-Shahar et al, 2008; Unal et al, 2008; Rowland et al, 2009; Sutani et al, 2009; Whelan et al, 2011), which acetylate SMC3 via distinct mechanisms

[1]Institute of Physiological Chemistry, Medical Faculty Carl Gustav Carus, Technische Universität Dresden, Dresden, Germany   [2]Department of Genes and Behaviour, Max Planck Institute for Biophysical Chemistry, Göttingen, Germany

Correspondence: rolf.jessberger@tu-dresden.de
François McNicoll's present address Institute of Cell Biology and Neuroscience, Wolfgang von Goethe Universität, Frankfurt am Main, Germany
*François McNicoll and Anne Kühnel contributed equally to this work

(Minamino et al, 2015). In mitotic cells, SMC3 acetylation by ESCO2 supports the recruitment of sororin, which antagonizes the anti-cohesion functions of other regulators (Nishiyama et al, 2010). During meiosis, cohesin localizes along chromosome axes and is essential for AE formation and SC assembly (reviewed in Jessberger (2012), Lee (2013), McNicoll et al (2013), Ishiguro (2019)). Anti-cohesion factors WAPL and PDS5B localize to AEs in mouse spermatocytes (Kuroda et al, 2005; Fukuda & Hoog, 2010), but their functions in spermatocytes have not yet been reported, which is true also for other cohesin regulators such as ESCO1 and ESCO2. Indeed, very little is known about the biological role of any cohesin regulatory factor in meiosis. This issue is very important not only for understanding basic spermatocyte and oocyte chromosome dynamics but also for deciphering processes and factors that determine the long-term stability of cohesin on meiotic chromosomes.

Here, we set out to analyze the role of the SMC3 acetyltransferase ESCO2 in male mouse meiosis and found that it is required for completion of SC formation between homologous chromosomes, that it contributes to the maintenance of sister chromatid cohesion on sex chromosomes, and that it is ultimately essential for spermatogenesis and male mouse fertility.

## Results

### Expression of ESCO2 and acSMC3 in spermatocytes

A critical step for cohesion establishment during mitotic S phase is the acetylation of the cohesin subunit SMC3 on two conserved lysine residues, K105 and K106 (hereafter acSMC3), by the nonredundant cohesin acetyltransferases ESCO1 and ESCO2 (Rolef Ben-Shahar et al, 2008; Unal et al, 2008; Rowland et al, 2009; Sutani et al, 2009; Whelan et al, 2011). In meiosis, cohesin subunit SMC1$\beta$ cannot be detected on chromosomes until early leptonema, that is, after the premeiotic S phase, and the protein is present throughout meiosis until metaphase II (Revenkova et al, 2001; Eijpe et al, 2003; Hodges et al, 2005). An antibody specifically recognizing acetylated acSMC3 readily co-immunoprecipitated acSMC3 with SMC1$\beta$ from mouse testis nuclear extracts (Fig 1A). SMC3 acetylation of SMC1$\beta$-containing cohesin complexes after premeiotic S phase would require expression of a cohesin acetyltransferase in spermatocytes.

Continuous mRNA expression in meiosis I suggested continued ESCO2 synthesis (Hogarth et al, 2011). To assess whether ESCO2 protein is expressed after premeiotic S phase, we sorted different cell populations from wild-type (wt) testes by FACS according to their DNA content and performed immunoblot analyses using an antibody specific for ESCO2 (Whelan et al, 2011). ESCO2 was clearly detectable in primary spermatocytes (4N cells) and spermatids (1N cells) (Fig 1B and C). Thus, unlike Esco2 mRNA levels, which are significantly lower in late meiotic and postmeiotic spermatocytes, the ESCO2 protein appears to be rather stable during meiosis and spermiogenesis. In contrast, in proliferating somatic cells, high ESCO2 levels are restricted to the S phase and dramatically decrease before the onset of mitosis (Hou & Zou, 2005).

### ESCO2 most likely acetylates SMC3 during SC formation

To determine whether cohesin acetylation might be involved in SC assembly, we performed immunofluorescence (IF) staining of mouse spermatocyte chromosome spreads. ESCO2, acSMC3, and sororin all clearly appeared on chromosomal AEs only upon synapsis in zygonema (Figs 1D–H, S1, S2, and S3), suggesting cohesion establishment at the time and location of SC assembly. Signals in leptonema may perhaps be very weak because of the more decondensed state of chromosomes at this stage, but short axes are already formed (Fig S1A). In zygonema, as homologous chromosomes begin to pair and synapse, acSMC3 intensity strongly increases on stretches of AEs that are already synapsed (Figs 1D and S1A and B). This increase was beyond the mere doubling of intensity that may have been due to the superimposition of synapsed AEs, suggesting a requirement for acSMC3-stabilized cohesion long after premeiotic DNA replication. In pachynema, when all autosomes are completely synapsed, acSMC3 localized along the SCs of autosomes and remained clearly enriched on AEs until their desynapsis in diplonema (Fig S1), indicating that cohesive cohesin complexes might be removed during SC disassembly. This is in line with recent suggestions of a prophase-like pathway-dependent removal of some cohesin before metaphase I (reviewed by Challa et al (2019)). Despite the fact that sex chromosomes display very high levels of total SMC3, acSMC3 was hardly detectable along their largely unsynapsed AEs (Fig 1E and F). Consistent with this, both ESCO2 and sororin were enriched along synapsed AEs of autosomes, whereas sex chromosomes showed only very weak IF signals for these two proteins (Figs 1G and H, S2, S3, and S4 for an antibody control; Fig S4 shows whole nucleus staining and shows that an antibody used in a previous study elsewhere [Evans et al, 2012] is not specific for ESCO2).

Together, these data suggest that SMC3 acetylation levels are high in synapsed regions, where they may support synapsis and/or be supported by synapsis. However, no enrichment of acSMC3 was observed at the PAR where the sex chromosomes are synapsed (Fig 1F, asterisk; see also below). Thus, although synapsis correlates with high levels of acSMC3 on autosomes, this is not the case on sex chromosomes. Sex chromosomes form a specific chromatin domain, a separate phase within the nucleus called "sex body," characterized by silencer marks such as $\gamma$H2AX (Fernandez-Capetillo et al, 2003; Handel, 2004). Because acSMC3 levels are low in both unsynapsed and synapsed parts of the X and Y chromosomes, it seems more likely that the particular sex body chromatin rather than asynapsis per se negatively impacts acSMC3 presence in this chromatin environment. This is consistent with loss of synapsis of autosomes, which are localized within the sex body chromatin of Esco2-deleted mice (see below).

To better visualize ESCO2 and acSMC3 along AEs of meiotic chromosomes, we used super-resolution structured illumination microscopy (SIM). SIM allowed visualization of both lateral elements of autosomal SCs, marked by the protein SYCP3, between which both acSMC3 and ESCO2 were clearly enriched (Fig 1I and K, white arrows), consistent with a possible role for cohesion establishment in SC formation and/or maintenance. SIM confirmed the very low levels of acSMC3 and ESCO2 on sex chromosome AEs, including in the PAR (Fig 1J and L, asterisk). Interestingly, SIM revealed that unsynapsed regions of sex chromosome AEs comprise two SYCP3-positive AEs (Fig 1J and L, yellow arrows). This is in agreement with earlier observations using electron microscopy, which suggested that AEs of both autosomes and sex chromosomes

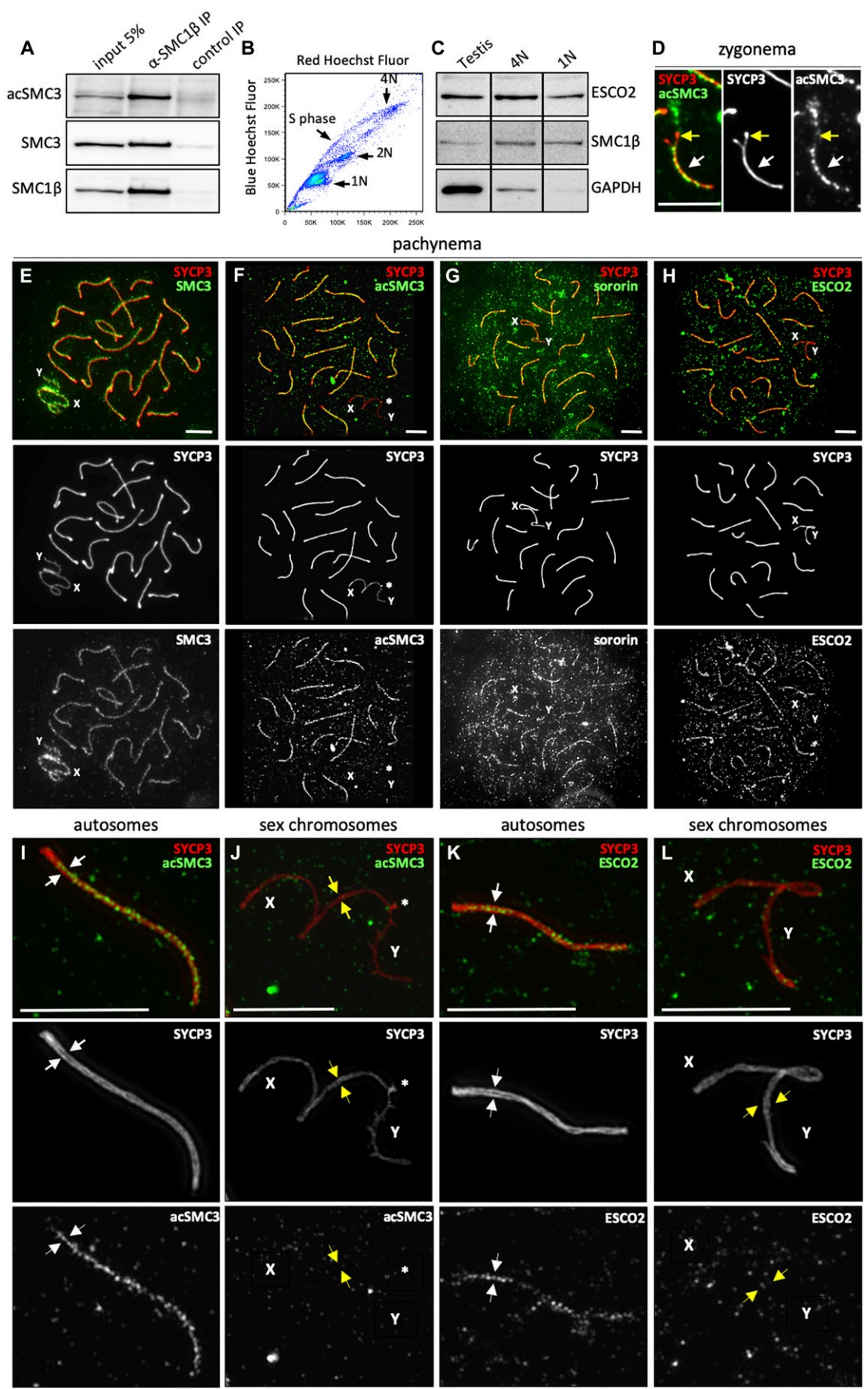

consist of two closely associated filaments, proposed to each represent one sister chromatid (Solari, 1969a; Solari, 1970; Solari & Tres, 1970; Chandley et al, 1984; del Mazo & Gil-Alberdi, 1986; Dietrich et al, 1992). Unsynapsed autosomal AEs in zygonema and diplonema did not appear as two strands at the resolution used here, which might indicate a stronger sister chromatid cohesion on autosomes than on sex chromosomes (see below). In line with this, neither ESCO2 nor acSMC3 appeared enriched between the sister chromatid AEs of sex chromosomes.

### *Esco2* is essential for male mouse gametogenesis

To determine the role of ESCO2 in meiosis, mice carrying a floxed *Esco2* allele (*Esco2^{fl}*, [Whelan et al, 2011]) were mated with mice expressing the CRE recombinase in germ cells shortly before (*Vasa-Cre* or *Stra8-Cre*) or after (*Smc1β-Cre*) the premeiotic S phase (Gallardo et al, 2007; Sadate-Ngatchou et al, 2008; Adelfalk et al, 2009). Efficient excision was confirmed by PCR analysis of FACS-sorted cells, primary spermatocytes (4N) and spermatids (1N), obtained from adult testes (Figs 2A, S5, and S6A) or total cells from testes of young males undergoing the first synchronized wave of meiosis (Fig S6B). We, thus, refer to spermatocytes of CRE-positive *Esco2^{fl/Δ}* or *Esco2^{fl/fl}* mice as *Esco2^{Δ/Δ}* spermatocytes and specify the Cre driver where appropriate. As our subsequent analyses showed, in both Cre strains, ESCO2 protein levels only slowly decreased and, thus, the phenotypes observed result from hypomorphic mutants (see below). In both *Esco2^{fl/Δ}* *Vasa-cre* and *Esco2^{fl/Δ}* *Smc1β-cre* mice, testis weight in adults was reduced by 20–30%, and testis sections revealed smaller tubules. In *Esco2^{fl/fl}* *Smc1β-cre* mice, excision was tested in the first synchronized wave of meiosis in young males and found to start at about day 7 postpartum (Fig S6B). This time point correlates with the early- to mid-leptotene stage as determined by immuno-histological analysis of testis sections of this mouse strain (Fig S6C), that is, to about the time of appearance of SYCP3 in the spermatocytes and, thus, at entry into meiosis. Most of the adult mice were sterile, others developed some sperm, and were subfertile, indicating some variability in the rate of ESCO2 protein loss. We also analyzed apoptosis in the testis tubules of these strains by staining for the

apoptosis-specific cleaved PARP variant (clPARP) and observed increased apoptosis and absence or strong reduction in numbers of elongated spermatids and mature sperm (Figs S7A–C, S6C, and Table S1; for details, see legend to Table S1). *Esco2^{fl/Δ}* *Vasa-cre* males were sterile and most *Esco2^{fl/Δ}* *Smc1β-cre* males were either sterile or subfertile.

Unlike seen in many known mutants impaired in prophase I chromosome behaviour (Turner et al, 2005), we did not observe complete or widespread arrest of spermatogenesis at tubular stage IV (mid-pachynema) or at metaphase. This is consistent with the largely proper synapsis of most autosomes in the *Esco2^{fl/Δ}* *Vasa-cre* spermatocytes (Fig 2B and see below). Many round spermatids were present, but the testis tubules contained very few elongated spermatids, and the mice were infertile. This suggests that ESCO2 plays an essential role in spermiogenesis, consistent with the continuously high levels of ESCO2 protein observed in postmeiotic wild-type cells (see Fig 1C). However, one cannot rule out that defects acquired during meiosis (see below) may result in later death, that is, apoptosis at the round-to-elongated spermatid transition.

In heterozygous breedings (*Esco2^{fl/+}* *Smc1β-cre* males with *Esco2^{fl/+}* females), about one-third of the progeny carried either one or two of the non-excised floxed alleles. This is in agreement with the PCR data, which even in the adult did not show complete excision in the total pool of cells analyzed (Fig S6A). However, in the sorted 4N meiosis I spermatocytes, a small fraction of the floxed allele had not been excised, and a band diagnostic for the floxed allele was also present, but relatively weak, in the 1N cells. Nevertheless, in each *Esco2^{fl/Δ}* *Smc1β-cre* mouse analyzed, there were chromosomal aberrations in at least a large fraction of the cells (see below), probably those cells where excision was complete.

To exclude that the other cohesin acetyltransferase, ESCO1, becomes up-regulated in *Esco2^{Δ/Δ}* cells, we performed quantitative RT–PCR on FACS-sorted c-kit–positive spermatogonia, 4N, 2N, and 1N cells from their testes and that of controls. The data show no significant increase in the expression of *Esco1* in the *Esco2^{Δ/Δ}* cells of both Cre drivers (Fig S8). In some populations, it appeared as if there is even a mild reduction in *Esco1* expression in the *Esco2^{Δ/Δ}* cells, but it was not statistically significant. Our own observations

**Figure 1. Expression and localization of cohesion establishment factors during the first meiotic prophase.**
**(A)** Meiotic cohesin complexes are acetylated on their SMC3 subunit. The meiosis-specific subunit SMC1β was immunoprecipitated from mouse testis nuclear extracts and acetylation of SMC3 was verified by immunoblotting using an anti-acSMC3 antibody. The membrane was then incubated with anti-SMC1β, stripped, and re-incubated with an anti-SMC3 antibody. **(B, C)** ESCO2 is expressed at high levels in meiotic and postmeiotic cells. **(B)** FACS profile of testis cells stained with Hoechst 33342; S phase, 1N, 2N, and 4N cell populations are indicated. The purity of the populations in these FASCS sorts was between 86% and 95%. **(C)** Immunoblotting of protein extracts from sorted primary spermatocytes (4N, $3 \times 10^5$ cells) and spermatids (1N, $1 \times 10^6$ cells) using anti-ESCO2, anti-SMC1β, and anti-GAPDH antibodies. GAPDH was used as a loading control, showing that ESCO2 and SMC1β are both enriched in meiotic and postmeiotic cells. **(D)** AcSMC3 appears in synapsed regions of homologous chromosomes during zygonema. Immunofluorescence staining of spermatocyte chromosome spreads using anti-SYCP3 (red) and anti-acSMC3 (green) antibodies. One pair of homologous chromosomes in zygonema is shown. A white arrow indicates the region where the homologs are already synapsed, and a yellow arrow marks the region where synapsis remains to be completed. **(E, F, G, H)** Cohesin and cohesion establishment factors are enriched in synapsed axial element (AE) regions during pachynema. Immunofluorescence staining of pachytene chromosomes using anti-SYCP3 (red) and either (green) anti-SMC3, anti-acSMC3, anti-sororin, or anti-ESCO2. These structures were visualized by conventional fluorescence microscopy and entire sets of chromosomes from single cells are shown. Sex chromosomes are labeled as X and Y. **(F)** Asterisk in (F) indicates the pseudoautosomal region. **(F, I, J)** The nucleus shown in (F) was visualized using super-resolution structured illumination microscopy (SIM), and the detailed structure of one pair of synapsed autosomes and of the sex chromosomes is shown. SIM allows visualization of the two lateral elements of the synaptonemal complex, which are marked by white arrows. The AEs of sex chromosomes, which remain unsynapsed, also appear as close, parallel double-filaments, that is, AEs corresponding to the individual sister chromatids are visible (yellow arrows) when visualized by SIM during pachynema (see Fig 2). **(K, L)** An asterisk depicts the short region of homology between chromosomes X and Y (pseudoautosomal region) (K, L). **(H, I, J)** Same as (I, J), except that the cell shown in (H) was analyzed. Note the enrichment of both acSMC3 and ESCO2 on autosomes between the lateral elements of the synaptonemal complex and the virtual absence of both proteins from sex chromosome AEs. A stretch of visibly separate sister chromatids is indicated by yellow arrows. Scale bars = 5 *μm*.

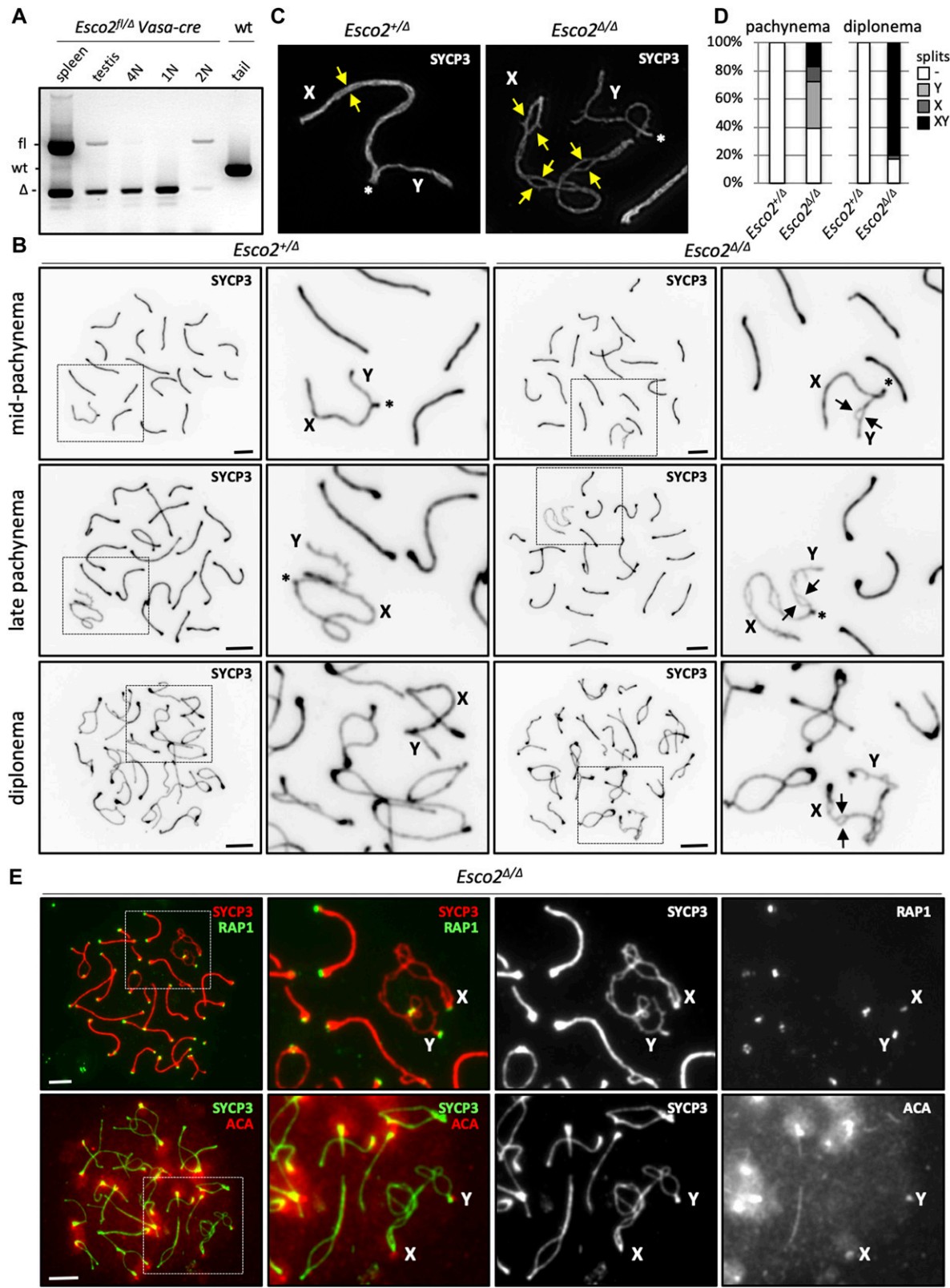

**Figure 2. Esco2 deletion during meiosis causes a loss of sister chromatid cohesion along sex chromosome arms.**
**(A)** Germ cell–specific deletion of the *Esco2* gene. Single-cell suspensions were prepared from testes of *Esco2^{fl/Δ} Vasa-cre* mice, stained with PI, and sorted by FACS according to the genomic DNA content. DNA was extracted from sorted cells and analyzed by PCR. The PCR reaction yields a 231-bp product for the wild-type (wt) allele, a 347-bp product for the floxed (fl) allele, and a 170-bp product for the excised (Δ) allele. DNA from the spleen of the same mice was used as control to verify whether excision

show that ESCO1-deficient mice are viable and fully fertile with no obvious meiotic phenotypes, and therefore, at least in a wild-type strain, ESCO1 has no major role in meiosis. Thus, although one cannot fully exclude that ESCO1 may partially compensate for ESCO2 activity, it obviously cannot rescue the phenotypes reported in this communication. Neither can it rescue embryonic lethality of the constitutive ESCO2 deficiency (Whelan et al, 2011).

### Esco2 maintains sister chromatid cohesion along sex chromosome arms during meiosis

IF staining of chromosome spreads from both *Esco2^{fl/Δ} Vasa-cre* and *Esco2^{fl/Δ} Smc1β-cre* spermatocytes using an antibody directed against SYCP3 revealed that sex chromosome AEs often separated into two distinct strands (hereafter "splits") over several regions along their length, indicating sister chromatid cohesion defects (Fig 2B). As mentioned above, previous studies using electron microscopy (Solari, 1969b, 1970; Solari & Tres, 1970; Tres, 1977; Goetz et al, 1984) and our analyses using SIM (Fig 1J and L) suggest that the AEs of sex chromosomes in wt spermatocytes each consist of two distinct SYCP3-positive filaments, each likely representing one sister chromatid. Mid- and late pachynema can be distinguished by the appearance of the X/Y chromosomes, which are more compact in late pachynema, by the more intense SYCP3 signals at the chromosome ends and by the appearance of histone H1t after mid-pachynema. A similar but less dynamic structure was described for autosomal AEs, where two filaments were proposed to each represent one sister chromatid, but those filaments remain in tight association throughout the meiotic prophase (del Mazo & Gil-Alberdi, 1986; Dietrich et al, 1992). Thus, the separation of sex chromosome AEs in *Esco2^{Δ/Δ}* spermatocytes suggests sister chromatid cohesion defects. Using super-resolution SIM, we confirmed that the AEs of chromosomes X and Y in wild-type spermatocytes each consist of two distinct SYCP3-positive filaments, each most certainly representing one sister chromatid (Figs 1J and L and 2C left). The very clear separation of both sex chromosome AEs each into two distinct strands in *Esco2^{Δ/Δ}* spermatocytes (Fig 2C, right) further supports this hypothesis. Thus, operationally, we defined true splits, that is, clearly abnormally separated SYCP3-positive sister chromatids, as those observed by conventional microscopy, which were not observed in wild-type or *Esco2^{+/Δ}* spermatocytes.

In mutant spermatocytes, splits were hardly observed in late zygonema and their occurrence increased with meiotic progression, from occasional small splits on the X and/or Y chromosome(s) around the mid-pachynema to multiple, larger splits on both X and Y chromosomes most of late pachytene and early diplotene spermatocytes (Fig 2B–D). The most straightforward interpretation of this phenotype suggests a defect in the maintenance or—if possible—in re-establishment of sister chromatid cohesion during meiosis. One cannot fully exclude other explanations such as a split of protein axes without the sister chromatids, but there is neither evidence for SYCP3-free axial sister chromatids nor for two axes of only protein. In wt and *Esco2^{+/Δ}* spermatocytes, SYCP3-positive sister chromatids became clearly visible by SIM around mid-pachynema but remained close together and parallel to each other (Fig 2C, left) and were less distinguishable in late pachynema and diplonema (Fig S9A). In *Esco2^{Δ/Δ}* spermatocytes, SYCP3-positive sister chromatids were much further separated from each other (see Fig 2B and C, right; Fig S9A), causing the large splits observed by conventional fluorescence microscopy. These data together with the above-described low abundance of acSMC3 on sex chromosomes suggest that a natural process causes a weakening of sister chromatid cohesion on unsynapsed regions of the sex chromosomes and that ESCO2 is required to maintain at least a low level of cohesion. The PAR appeared unaffected, suggesting that synapsis supports cohesion maintenance (see also below). Using the *Vasa-cre* driver, we cannot exclude that potential defects in sister chromatid cohesion establishment during premeiotic S phase could contribute to this later phenotype. However, centromere numbers determined by ACA staining in leptonema were the same in mutant and wt at 39.1 (±3.9; n = 57) versus 38.6 (±2.7; n = 50) per cell, indicating that centromeric cohesion is not significantly deficient in both strains. Also, given that the sex chromosomes become morphologically and dynamically (pairing) very distinct from autosomes only during prophase I, this should explain the chromosome-specific phenotype seen here. The use of *Smc1β-iCre* confirmed that this phenotype is not a consequence of premeiotic excision (Figs S7 and S10). Splits were seen on about 24% of X and Y chromosomes in *Esco2^{Δ/fl}* mice positive for *Smc1β-iCre* (24.1% ± 4.1%; n = 679 cells), whereas very few splits (1.1% ± 0.8%; *P* [wt versus mutant] = 0.015; n = 608 cells) were seen in the *Esco2^{Δ/+} Smc1β–iCre* littermates. In cells where splits appeared, the extent of splits was comparable with that seen in *Esco2^{fl/Δ} Vasa-cre* mice. This strongly suggests that there is no or only a very minimal effect of presumed premeiotic excision of *Esco2* in the *Vasa-Cre* strain (see also below and Fig S11).

We also compared frequencies and lengths of the splits seen in the *Esco2^{Δ/Δ}* spermatocyte sex chromosomes between cells derived from the two Cre drivers (Fig S11). The total lengths of X and Y

---

is germ cell specific; DNA from the tail of wt mice was used to show the wt allele. See also Fig S5. **(B, C, D, E)** *Esco2* deletion in spermatocytes causes sister chromatid cohesion defects along sex chromosome arms. **(B)** Immunofluorescence (IF) staining of spermatocyte chromosome spreads from control (*Esco2^{+/Δ}*) and *Esco2^{fl/Δ} Vasa-cre* mice (*Esco2^{Δ/Δ}* spermatocytes) at different stages of meiotic prophase using an anti-SYCP3 antibody and visualized by conventional fluorescence microscopy. Representative images of each stage are shown. Sex chromosomes are labeled with X and Y in magnified images, and the pseudoautosomal region, where visible, is indicated by an asterisk. Examples of sister chromatid cohesion defects such as splits in axial elements are indicated by arrows. **(C)** Sex chromosome axial elements consist of two SYCP3-positive filaments, presumably one per sister chromatid. Super-resolution structured illumination microscopy was performed on pachytene chromosome spreads from *Esco2^{+/Δ}* and *Esco2^{Δ/Δ}* spermatocytes after IF staining with an anti-SYCP3 antibody. The pseudoautosomal region is indicated by an asterisk, and visible individual strands (in the control) or examples of clear splits (in the *Esco2^{Δ/Δ}*) are marked by yellow arrows. **(D)** Percentage of pachytene and diplotene spermatocytes showing no splits (-), splits on chromosome Y only (Y), on chromosome X only (X) or on both sex chromosomes (XY) when visualized by conventional fluorescence microscopy. >100 cells of each genotype where X and Y could be clearly identified were analyzed. **(E)** *Esco2* deletion during meiosis does not affect centromeric and telomeric cohesion. IF staining of chromosome spreads from *Esco2^{Δ/Δ}* spermatocytes using different combinations of antibodies: *top row*, anti-SYCP3 (red) and anti-RAP1 (telomere-binding protein, green); *bottom row*, anti-SYCP3 (green) and anti-centromere (ACA, red), >80 cells were analyzed for ACA staining, including >40 cells in late pachynema. Scale bars for all images = 5 μm.

chromosomes was 22.82 and 24.81 μm for the *Vasa-Cre* and the *Smc1β-iCre* strains, respectively, and thus not different between the two strains (*P* > 0.1). The total lengths of all splits on these X and Y chromosomes per chromosome was 5.3 (±2.5, SD) and 3.7 (±4.3 SD) μm for the *Vasa-Cre* and the *Smc1β-iCre* strains and thus slightly lower for the latter (*P* = 0.02) although the spread was larger. The *Smc1β-iCre* strain also had fewer splits with a median of two per sex chromosome, whereas the *Vasa-Cre* cells showed approximately three splits per sex chromosome.

These data for *Esco2*$^{fl/Δ}$ *Vasa-cre* and *Esco2*$^{fl/Δ}$ *Smc1β-iCre* spermatocytes were indistinguishable from additional data obtained using a third model in some key experiments, *Esco2*$^{fl/Δ}$ *Stra8-cre*, where *Cre* is expressed shortly before entry into meiosis, that is, around day 3 pp (Sadate-Ngatchou et al, 2008) (Fig S12). Testis weight of *Esco2*$^{fl/Δ}$ *Stra8-cre* mice was reduced by 36% compared with controls. Of 239 pachytene *Esco2*$^{fl/Δ}$ *Stra8-cre* cells, 52% showed at least one clear split of sex chromosome axes, whereas the wt control showed none (263 cells counted; each from two mice). Clearly visible splits on sex chromosomes and asynapsis and axis splits of autosomes that reach into the sex body chromatin were observed like in the other two models.

Although extensive separation was often observed in *Esco2*$^{Δ/Δ}$ spermatocytes independently of the Cre driver used along sex chromosome arms, sister centromeres and telomeres of sex chromosomes always remained in tight association (>80 cells analyzed, Fig 2E). In late diplonema, when sex chromosome AEs are normally shortened and thickened (Page et al, 2006), splits became less frequent, and in many cells, they were not visible at all (Fig S9). Thus, the axis splits are transient. Because there is some cell death during meiotic prophase, particularly in pachynema, this apparent recovery in diplonema may be due to re-association of splits and/or selection against spermatocytes that still carry splits. The critical role of ESCO2 identified here in supporting sex chromosome sister chromatid cohesion is transient and limited to pachynema and early diplonema. Because in pachynema increased apoptosis was observed, the true extent of splits may be even larger and most dramatic in the cells that were to die already in pachynema.

## Cohesin subunit association with sex chromosomes is affected by *Esco2* deficiency

As noted above, in wt and *Esco2*$^{+/Δ}$ spermatocytes, cohesion appears transiently weakened along sex chromosome AEs around mid-pachynema. Concomitantly with the tight re-association of SYCP3-positive sister chromatids in late pachynema and diplonema, cohesin is enriched on sex chromosomes (Fig 1E). The SMC3 subunit is present in all meiotic cohesin complexes and, thus, provides a proper indication of total cohesin levels. In *Esco2*$^{Δ/Δ}$ spermatocytes, SMC3 levels on sex chromosome AEs were similarly high to those in wt spermatocytes, and cohesin was abundant on either side of the splits in the mutant (Fig 3A and B). This contrasts the very low levels of acSMC3 on the sex chromosomes of both wt and *Esco2*$^{Δ/Δ}$ pachytene spermatocytes (Fig 1 and see also Fig S17C). One possible explanation for these observations is that most of cohesin on sex chromosome AEs may be rather weakly associated and not contributing to stable cohesion.

Similarly to total cohesin, RAD21 and RAD21L kleisin subunits, representing two distinct groups of cohesin complexes, are both enriched on wt sex chromosome AEs compared with autosome AEs in late pachynema and early diplonema (Fig 3C–F; also described in Ishiguro et al (2011)). RAD21 signals were generally more widespread, that is, axes-associated and also distributed throughout the surrounding chromatin. In *Esco2*$^{Δ/Δ}$ pachytene spermatocytes, the staining pattern of RAD21 was similar to that of total cohesin, that is, it was present on either side of the splits, and RAD21 levels on sex chromosomes were similar to those of control (*Esco2*$^{+/Δ}$) spermatocytes (Figs 3C and D and S13). This suggests that cohesin complexes containing RAD21 are retained on sex chromosomes despite a loss of cohesion, consistent with a non-cohesive role of RAD21 complexes in meiosis.

By contrast, RAD21L was largely absent from AE regions where sister chromatid cohesion was impaired, that is, splits (Figs 3E and F and S14), indicating that some of the RAD21L cohesin complexes were lost from sex chromosomes because of the lower dosage of ESCO2. This loss argues for the presence of at least a small amount of acSMC3 cohesin complexes on wt sex chromosomes. The continuous staining pattern observed for RAD21L along sex chromosome AEs in zygonema and early pachynema, that is, before the appearance of splits suggests defects in maintaining RAD21L levels with progressing chromatid splitting. This also suggests a role for RAD21L in cohesion, at least on the sex chromosomes.

The major cohesion-mediating kleisin REC8 was present on wt and mutant pachytene chromosomes, including the X and Y chromosomes, although its levels on sex chromosomes and in synapsis-defective autosomal regions were lower than on synapsed autosomes (Figs 3G–J and S15). REC8 signals are seen at different locations in different spermatocytes. Although we did rarely observe REC8 signals on the two SYCP3-stained chromatids within split regions of sex chromosomes, we occasionally found REC8 signals accumulated on one of the sister chromatids in split regions (Fig S16). This may suggest that in the sex body, upon loss of cohesion REC8 can stay associated with one sister chromatid.

## ESCO2 hypomorphism causes spermatogenesis defects

Despite the clearly observed phenotypes, we measured only a very moderate decrease in ESCO2 IF signal intensity in *Esco2*$^{Δ/Δ}$ spermatocytes compared with their *Esco2*$^{+/Δ}$ controls (Fig S17A, B, and E). Accordingly, IF signals for acSMC3 did not suggest a quantitative difference (Fig S17C), nor did the levels of acSMC3 that co-immunoprecipitated with SMC1β (Fig S17D). Although CRE-mediated recombination was highly efficient (Fig 2A), it appears that this leads to only a partial depletion of ESCO2 protein in spermatocytes and thus to ESCO2 hypomorphism, presumably as a result of the high stability of this protein as indicated above (see Fig 1C). Given that sex chromosome AEs harbor less ESCO2, acSMC3, and sororin than autosomal AEs (Fig 1F–L), they might be most sensitive to such a modest dosage effect. Control animals of either wt or fl/+ genotypes expressing CRE from either of the three *Cre* driver strains did not show any phenotype.

Because gametogenesis is abrogated at the round-to-elongated spermatid transition, we also stained round spermatids of *Esco2*$^{Δ/Δ}$

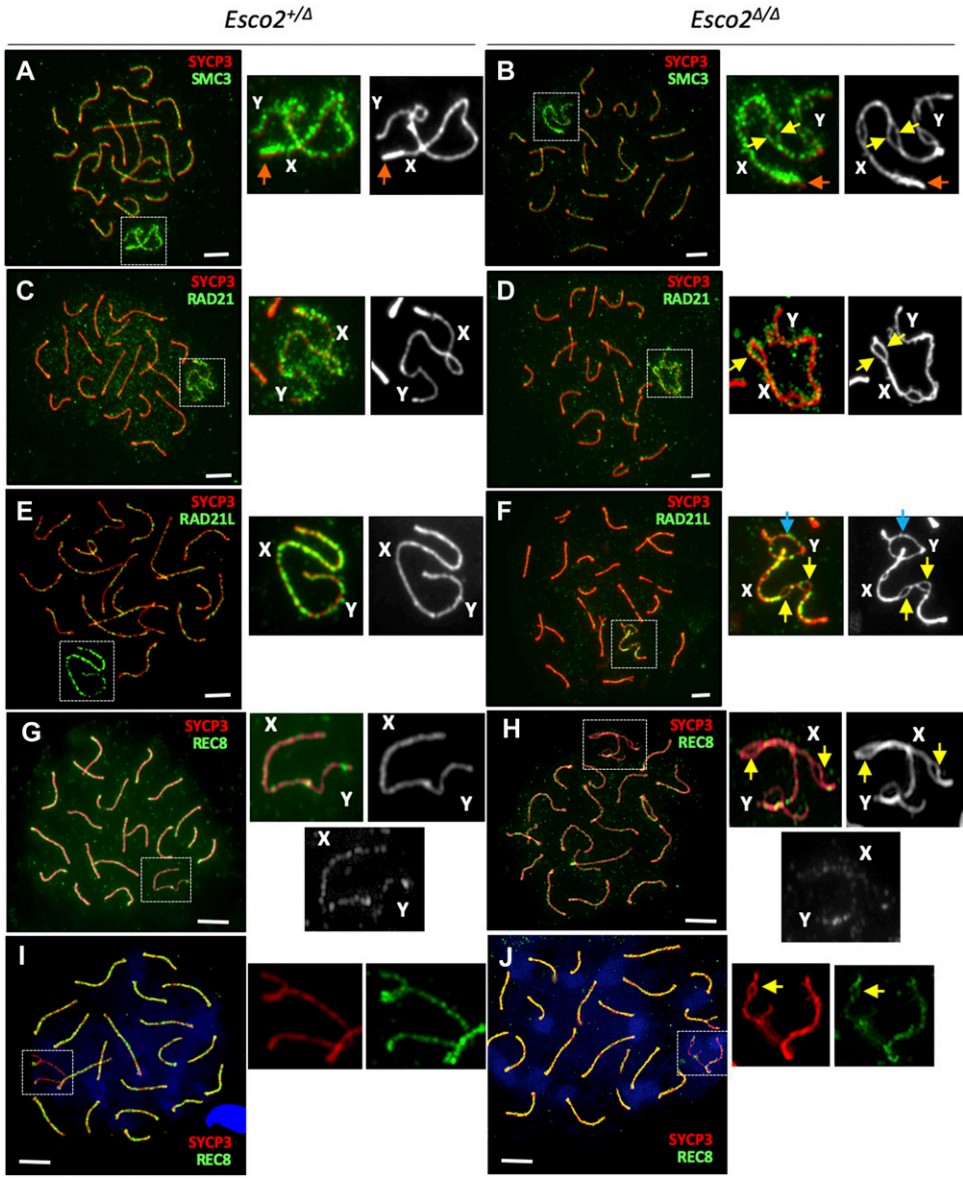

Esco2^{+/Δ}                    Esco2^{Δ/Δ}

**Figure 3. Cohesin enrichment on sex chromosomes and changes upon *Esco2* deletion.**
**(A, B, C, D, E, F, G, H, I, J)** Immunofluorescence staining of chromosome spreads from control (*Esco2^{+/Δ}*; A, E and G, I and *Esco2^{Δ/Δ}*; B, F and H, J) spermatocytes using different combinations of antibodies: (A, B) anti-SYCP3 (red) and anti-SMC3 (subunit present in all cohesin complexes, green), showing the enrichment of cohesin on sex chromosomes compared with autosomes in both *Esco2^{+/Δ}* and *Esco2^{Δ/Δ}* spermatocytes. Magnified images on the right show the immunofluorescence signals on sex chromosomes obtained with both antibodies (colored) or with SYCP3 only (black and white). **(A, B, C, D, E, F, G, H, I, J)** Note the enrichment of cohesin at the centromere of the X chromosome in both *Esco2^{+/Δ}* and *Esco2^{Δ/Δ}* spermatocytes (orange arrows) and its presence on either side of the splits in *Esco2^{Δ/Δ}* spermatocytes (yellow arrows); (C, D) anti-SYCP3 (red) and anti-RAD21 (cohesin subunit present in a subset of cohesin complexes, green), showing a similar staining pattern as total cohesin in (A, B); (E, F) anti-SYCP3 (red) and anti-RAD21L (meiosis-specific cohesin subunit, green), showing the enrichment of RAD21L on sex chromosomes compared with autosomes in *Esco2^{+/Δ}* spermatocytes and the relatively low abundance of RAD21L in regions where sister chromatid cohesion is impaired in *Esco2^{Δ/Δ}* spermatocytes; (G, H, I, J) anti-SYCP3 (red) and anti-REC8 (meiosis-specific cohesin subunit, green), showing no particular enrichment of REC8 on sex chromosomes. **(G, H, I, J)** For (G, H) and (I, J), two independent anti-REC8 antibodies were used. **(F)** The blue arrow marks a synapsis-defective autosomal region with a sex chromosome-like morphology (F). "A" indicates autosomes, "X" and "Y" indicate the sex chromosomes; scale bars = 5 *μm*.

and *Esco2^{+/Δ}* mice, both *Cre*-positive, for ESCO2 and acSMC3 (Fig S18). Both proteins were detectable in round spermatids of *Esco2^{Δ/Δ}* and *Esco2^{+/Δ}* mice, regardless whether the *Vasa-cre* or *Smc1β-iCre* was used. Quantification showed a mild, barely statistically significant reduction for ESCO2 by 1.2-fold in the *Vasa-Cre Esco2^{fl/Δ}* strain. A clearly significant reduction was seen in this strain for acSMC3, whose levels were 2.3-fold decreased. Both ESCO2 and acSMC3 levels were significantly different between *Smc1β-iCre Esco2^{Δ/Δ}* and *Esco2^{+/Δ}* spermatids. ESCO2 levels were about threefold lower and acSMC3 levels were reduced by approximately eightfold in *Smc1β-iCre Esco2^{Δ/Δ}* spermatids. This shows a more severe effect of the *Esco2* deletion at the round spermatid stage and thus may cause the death of round spermatids. It also indicates relatively high stability of the protein and its gradual disappearance with progression of meiosis in our mouse models. This further points to an interesting role of ESCO2 at this spermatid stage, which

shall be subject to future spermiogenesis studies. In agreement with the notion above on the hypomorphic meiotic phenotype, this further suggests that it indeed takes many days after excision of the gene until ESCO2 protein levels finally start to significantly decrease. Some cells in which ESCO2 levels may have decreased faster may have died in pachynema as indicated by the increase in apoptosis at that stage. At any rate, even the mild decrease during prophase I suffices to trigger partial loss of sister chromatid cohesion on the sex chromosomes.

## ESCO2 promotes SC formation

Meiotic cohesin is required for proper synapsis between homologous autosomal chromosomes and for synapsis at the PAR of sex chromosomes as inferred from several cohesin deficiency models (reviewed in Lee (2013), McNicoll et al (2013)). Synapsis is essential

for meiotic progression. Various synapsis-defective mutant sper-matocytes, including cohesin mutants, are eliminated at stage IV of the testicular epithelial cycle by the so-called mid-pachytene checkpoint (reviewed in Turner et al (2005), Burgoyne et al (2009)). To determine whether synapsis is affected by deletion of the *Esco2$^{fl}$* locus, we co-stained spermatocyte chromosome spreads for SYCP3, the asynapsis marker γH2AX and the testis-specific histone variant H1t, which is expressed only after mid-pachynema (Drabent et al, 1996) (Fig S19A). Compared with *Esco2$^{Δ/+}$* controls, a much higher percentage of the *Esco2$^{Δ/Δ}$* spermatocytes displayed a chromo-somal configuration corresponding to leptonema or zygonema, suggesting a synapsis delay in *Esco2$^{Δ/Δ}$* spermatocytes (6% of wt versus 34% of *Esco2$^{Δ/Δ}$*, 200 cells counted per genotype). In many *Esco2$^{Δ/Δ}$* spermatocytes, a small number of autosomes (typically 1–3 per cell) remained partly unsynapsed throughout pachynema and diplonema (Fig 4A). Interestingly, the unsynapsed portion of these autosomes was always in close association with the sex chromosomes, that is, appeared to be embedded in the γH2AX-positive sex body chromatin (Fig 4A). Both sex chromosomes and associated synapsis-defective autosomes in *Esco2$^{Δ/Δ}$* spermato-cytes were efficiently silenced as revealed by staining for elon-gating (phosphoS2) RNA polymerase II (Fig S19B), which does neither localize to the sex body in wt spermatocytes nor in *Esco2$^{Δ/Δ}$* spermatocytes.

In spermatocytes, centromeres are the last chromosomal re-gions to synapse and, together with nascent chiasmata, are the last sites to desynapse (Qiao et al, 2012). Most cases of autosomal asynapsis observed after mid-pachynema in *Esco2$^{Δ/Δ}$* spermato-cytes occurred at centromeric ends (92%, >75 synapsis-defective autosomes analyzed, Fig S19C), indicating that those autosomes failed to complete synapsis rather than undergoing premature desynapsis. Similar proportions of H1t-positive cells were found in *Esco2$^{Δ/Δ}$* and *Esco2$^{Δ/+}$* mice (Fig S19A), suggesting that partial asynapsis of up to three autosomes is not sufficient to trigger the mid-pachytene checkpoint related to failure of silencing of the sex chromsomes. Indeed silencing occurs as shown above in *Esco2$^{Δ/Δ}$* spermatocytes. The mild increase in pachytene apoptosis, seen in tubules of various stages, did not suffice to affect total post-pachytene cell numbers.

During mammalian meiosis, DSB repair facilitates homolog pairing and synapsis (Baudat et al, 2013). ESCO2 is required for DSB repair in somatic cells (Whelan et al, 2011), probably through genome-wide cohesion reinforcement (Strom et al, 2007; Unal et al, 2007). No roles have been reported for ESCO2 in meiotic DSB repair, where recombination usually occurs between homologous chro-mosomes rather than between sister chromatids. DSB foci marked by the meiosis-specific recombinase DMC1 appeared to be pro-cessed normally in *Esco2$^{Δ/Δ}$* spermatocytes. DSB foci were retained neither on synapsis-defective autosomes nor on sex chromosomes in late pachynema (Fig S20A and B). To test whether the initial numbers of DMC1 foci were different, which would be indicative of an early effect of *Esco2* deletion, we also determined the number of DMC1 foci in leptonema and zygonema. In leptonema we counted 20.6 foci per *Esco2$^{fl/Δ}$ Vasa-cre−* control cell (±25.8 SD; n = 19) and 38.5 foci per *Esco2$^{Δ/Δ}$ Vasa-cre+* cell (±38.0 SD; n = 37), a difference that was barely significant (*P* = 0.04). In zygonema we counted 106.9 foci (±45.3 SD; n = 30) per *Esco2$^{fl/Δ}$ Vasa-cre−* control cell and 107.4

foci (±41.7 SD; n = 32) per *Esco2$^{Δ/Δ}$ Vasa-cre+* cell. Thus, generation and processing of DMC1 foci appear largely unaffected by *Esco2* deletion. In about 25% of the *Esco2*-deleted cells we observed DMC1 foci also within the split regions of the sex chromosomes (examples with and without foci in split regions see Fig S20A, small insets), and occasionally on the rare split regions of autosomes that are close to the sex chromosomes. Notably, the presence of RAD51 or DMC1 foci in split regions strongly suggests that the axes seen in the splits contain DNA besides the SYCP3 protein and thus indeed are sister chromatids. Considering these data, it appears very unlikely that the synapsis defects reported above were due to a delay in meiotic DSB repair. The presence of RAD51 and DMC1 foci on unsynapsed AEs of the sex chromosomes, which generally repair their DSBs later than autosomes, may also suggest inter-sister recombination, similar to observations in REC8 deficient mice (Bannister et al, 2004), consistent with aberrant SYCP1 deposition between sister chromatids (see below). We also noticed in mid-pachytene that there were many more DMC1 foci on the few autosomal regions associated with the sex body than on autosomes elsewhere (Fig S20A). This suggests a sex body chromatin-mediated delay of au-tosomal DSB repair like for the sex chromosome themselves.

There was a tendency, although statistically not significant, of premature separation of the sex chromosomes from each other in diplotene: 36% of control cells (*Esco2$^{fl/+}$ Vasa-cre*; n = 19) and 65% of the *Esco2$^{fl/Δ}$ Vasa-cre* cells (n = 31) showed separated X and Y chromosomes (*P*-value = 0.051), possibly indicating slight weak-ening of PAR association.

Taken together, the above data suggest that ESCO2 plays a sup-portive role in SC assembly.

### A "cohesion-first" model for support of synapsis

As described above, in wt mouse spermatocytes, unsynapsed re-gions of sex chromosomes appear as two closely associated yet distinct filaments around mid-pachynema, as if these chromo-somes featured weaker cohesion or even had a natural tendency to locally lose cohesion, at least transiently. In *Esco2$^{Δ/Δ}$* spermato-cytes, the separation of the two SYCP3-positive sister chromatids is drastically exacerbated and continues through late pachynema and diplonema (see Figs 2B and C and S9). This suggests that asynapsis or other factors specific for the sex chromosome chro-matin around mid-pachynema cause(s) a weakening of sister chromatid cohesion and that ESCO2 is one of the factors required for maintaining at least a low level of cohesion. On the other hand, the autosomal synapsis delay in *Esco2$^{Δ/Δ}$* spermatocytes described above suggests that in this model, some cohesion defects actually occur before mid-pachynema and argues for a "cohesion-first" mode where cohesion supports synapsis. The fact that all reported mutant mouse meiocytes deficient for a particular cohesin subunit show various levels of asynapsis also argues for a supportive role of cohesin in synapsis (Bannister et al, 2004; Revenkova et al, 2004; Xu et al, 2005; Llano, Gomez et al, 2008, 2014; Herran et al, 2011; Fukuda et al, 2014; Hopkins et al, 2014; Winters et al, 2014).

To better define the timing of appearance of cohesion defects, we visualized synapsis-defective autosomes of *Esco2$^{Δ/Δ}$* spermato-cytes by SIM (Fig 4B). In early/mid-pachynema—the earliest stage at which it is technically possible to identify synapsis-defective

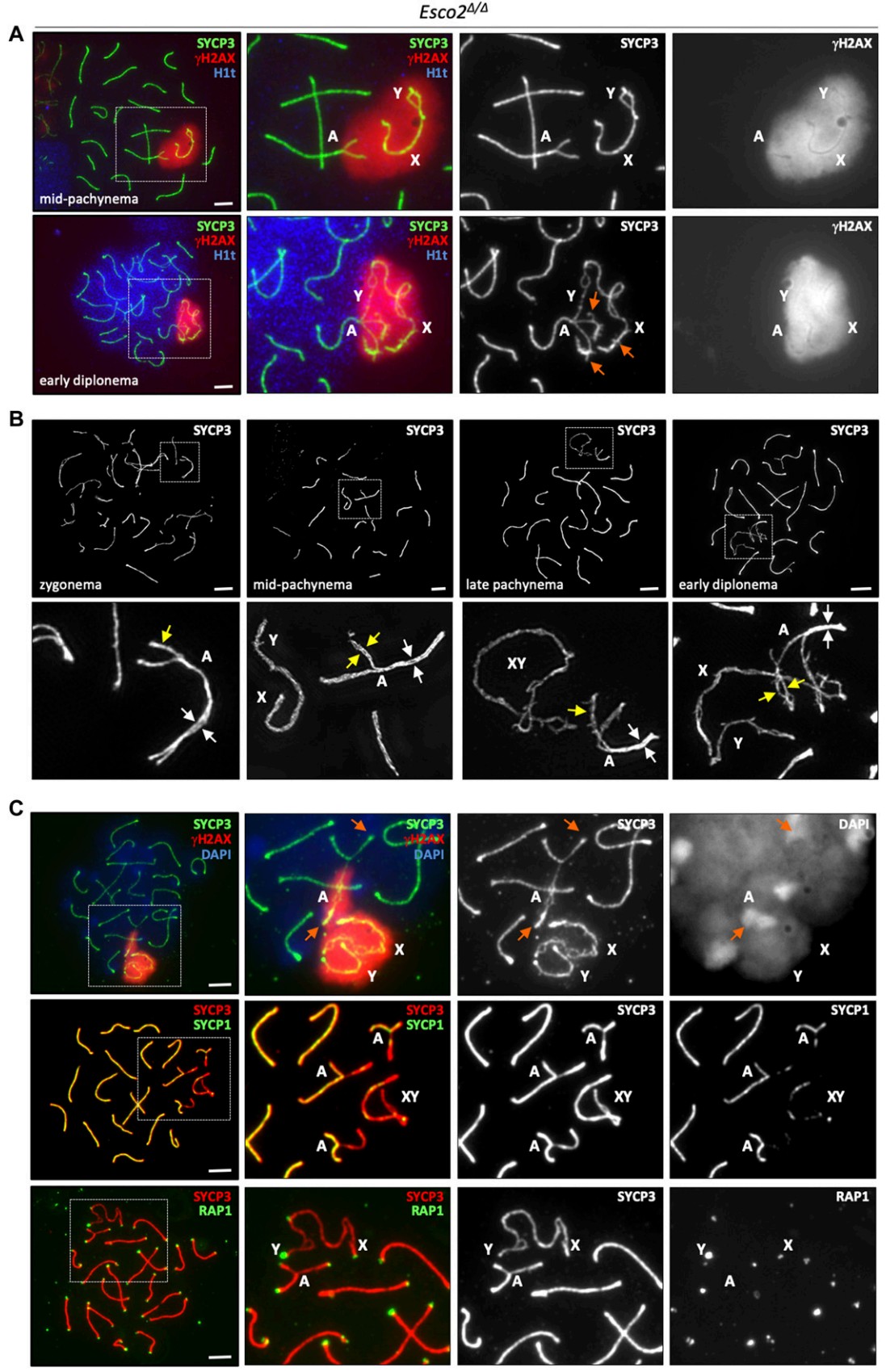

autosomes—individual SYCP3-positive sister chromatids were already clearly visible in the unsynapsed regions of autosomes close to sex chromosomes (compare unsynapsed versus synapsed regions of the same homolog pair in Fig 4B, marked by yellow and white arrows, respectively). Because these unsynapsed autosomal regions were always found within the sex body, this suggests that a weakening of sister chromatid cohesion caused by the sex body chromatin triggers asynapsis during pachynema. Thus, we propose a "cohesion first with later support by synapsis" model. In late pachynema and diplonema, the extent of separation of sister AEs was much greater, similar to that of sex chromosomes. Together, with the results presented above, where ESCO2, acSMC3 and sororin were all less abundant on unsynapsed sex chromosome AEs, these results suggest that cohesion maintenance through continuous acetylation is necessary for normal progression of meiotic prophase.

Synapsis-defective autosomes in $Esco2^{\Delta/\Delta}$ spermatocytes were always associated with the sex body and acquired a morphology typical of sex chromosomes by late pachynema and diplonema; their AEs were elongated and showed a characteristic thickening at centromeres and often many excrescences or hair-like structures (Fig 4A and B). Staining for the telomere-binding protein RAP1 ruled out the possibility that those were additional sex chromosomes because of aneuploidy (Fig 4C). In many cases, only one of the two autosome homologs had a sex chromosome-like appearance in its unsynapsed region, whereas the other one, outside the sex body, had the normal appearance of an autosome. The homolog with a normal appearance was positive for the SC protein SYCP1 and negative for the asynapsis marker γH2AX despite being unsynapsed; therefore, SC proteins within the unsynapsed region of such autosomes were probably deposited between its sister chromatids (Fig 4C). Such aberrant inter-sister SYCP1 deposition was described in REC8 deficient spermatocytes (Bannister et al, 2004; Xu et al, 2005) and in a recent study from our laboratory (Biswas et al, 2018). Splits were often observed in γH2AX-positive but not in γH2AX-negative regions, that is, not in regions with inter-sister synapsis-like SYCP1 accumulation (see example in Fig 4C). Thus, it appears that, similarly to inter-homolog synapsis, inter-sister deposition of SYCP1 and absence of γH2AX correlates with sister chromatid cohesion and prevention of chromosomes to acquire a sex chromosome-like morphology. This is in agreement with the suggestion that the sex body chromatin, which features γH2AX, impairs stable cohesion.

The data presented here, where cohesion is compromised through reduced ESCO2 levels, suggest that not merely cohesin proteins—which act in various processes—but rather acetylation-enforced cohesion mediated by cohesive cohesin supports synapsis.

## Discussion

Our study uncovered a particular weakness of sister chromatid cohesion in unsynapsed regions of male sex chromosomes during meiotic prophase. In mouse spermatocytes, sex chromosome cohesion appears to be kept in balance by the cohesion establishment factor ESCO2. In agreement with reports from other organisms (Severson et al, 2009; Weng et al, 2014), we hypothesize that cohesion maintenance is required not only during the premeiotic S phase but also during the meiotic prophase I. This hypothesis is based on our findings that (i) ESCO2 is present throughout meiosis; (ii) ESCO2, acSMC3, and sororin are abundant in synapsed autosomal regions and separated sister chromatids seen in $Esco2^{\Delta/\Delta}$ spermatocytes appear only in unsynapsed regions; (iii) a mildly reduced dosage of ESCO2 causes a phenotype specific for the sex chromosomes, which become morphologically and dynamically distinct only in meiosis and not before; (iv) the splits, separated sister chromatids, are transient and appear only in pachynema, not in the earlier stages of meiosis, where all or most chromosomes are unsynapsed. A problem in cohesion generated during the premeiotic S phase would very likely affect the earlier meiotic stages as well; (v) because essentially the same phenotypes were seen using the *Vasa-Cre, Stra8-Cre*, and the later expressing *Smc1β-iCre* drivers, this strongly supports a post-replicative role of ESCO2 in meiosis. This suggests that some cohesin complexes are cohesion-reinforced—if not newly loaded—during prophase I, supported by ESCO2; (vii) SMC1β/SMC3 complexes, which appear after entry into meiosis, are acetylated on their SMC3 subunit. This suggests either acetylation within early prophase I to convert non-cohesive cohesin complexes to cohesive ones, or usage of SMC3 that was acetylated earlier, before entry into meiosis.

In *Esco2*-deleted spermatocytes, the splits happen near the unsynapsed ends of the X and Y chromosomes, internally, of close to the PAR. There can be one split per chromosome, or there can be two or more. Sometimes, the splits are large, sometimes they are small. Thus, there is no specific location of these chromosomes where the splits are to happen. Rather, there is a general cohesion weakness of the unsynapsed X and Y chromosomes which randomly causes loss of cohesion along their axes.

---

**Figure 4. Sister chromatid cohesion supports synapsis during meiotic prophase.**
**(A)** *Esco2* promotes synaptonemal complex formation. Immunofluorescence staining of chromosome spreads from $Esco2^{\Delta/\Delta}$ spermatocytes using anti-SYCP3 (green), anti-γH2AX (marks unsynapsed chromatin, red), and anti-H1t (testis-specific histone variant demarcating the mid-pachynema onset, blue) antibodies. **(A)** *Top row*, example of cell with incomplete synapsis on one pair of autosomes (indicated by (A) in magnified image) around mid-pachynema, where spermatocyte nuclei are slightly H1t positive. **(A)** *Bottom row*, example of cell with one synapsis-defective autosome (γH2AX-positive, indicated by (A)) at early diplonema. Note the similar morphology of the synapsis-defective autosome axial element (AE) regions and of sex chromosome AEs; orange arrows indicate excrescences in proximity to the thickened centromeres. **(B)** Sister chromatid cohesion is weakened around mid-pachynema in synapsis-defective regions of $Esco2^{\Delta/\Delta}$ spermatocytes and is partly lost in early diplonema. Super-resolution structured illumination microscopy showing the structure of synapsis-defective autosomes stained with anti-SYCP3 at different stages. Yellow arrows indicate unsynapsed regions of individual homologs and white arrows indicate synapsed regions. Note that sister AEs are not visible in unsynapsed regions during zygonema but can be clearly distinguished around mid-pachynema and further separate from each other in early diplonema. **(C)** Asynaptic autosomes acquire a sex chromosome-like morphology in $Esco2^{\Delta/\Delta}$ spermatocytes. Immunofluorescence staining using different combinations of antibodies: *top row*, anti-SYCP3 (green) and anti-γH2AX (red), nucleic acids were stained with DAPI to show pericentric heterochromatin and centromeres of a synapsis-defective autosome pair are indicated by orange arrows; *middle row*, anti-SYCP3 (red) and anti-SYCP1 (transverse filament of the synaptonemal complex, green), where synapsed AE regions appear yellow in the merged image; *bottom row*, anti-SYCP3 (red) and anti-RAP1 (green). Scale bars for all images = 5 μm.

Male sex chromosomes are different from autosomes, for they show very little acSMC3, whether in the unsynapsed regions or in the synapsed PAR. Thus, cohesion is likely weaker on X and Y, in accordance with the limited separation of sex chromosome sister chromatids observed by SIM even in wt spermatocytes. The decreased ESCO2 levels render sex chromosomes particularly prone to loss of cohesion, yielding the prominent split phenotype reported here. It remains to be determined why ESCO2 is heavily underrepresented on sex chromosomes, and we speculate that this is caused by the particular features of the sex chromosome chromatin. This low level of ESCO2 may indeed contribute to the weakness of sex chromosome cohesion. In contrast to autosomes, which become transcriptionally active upon synapsis, the sex chromosomes are continuously silenced, and their chromatin is modified accordingly. This may impair the maintenance and/or de novo acetylation of SMC3. The observation in mutant spermatocytes that those autosomes showing loss of synapsis are associated with the sex body is consistent with the notion that particular features of the sex body chromatin determine acSMC3-related properties. The absence of enrichment of acSMC3 and ESCO2 also in the synapsed PAR of wt spermatocytes suggests that acSMC3 enrichment is not synapsis-dependent but rather prevented by the sex chromosome chromatin.

Lowered levels of ESCO2 also cause the loss of some RAD21L-containing cohesin complexes from regions of sex chromosome AEs that fell apart into splits. We also frequently observed strong REC8 signals on only one of the sister chromatids in split regions. Thus, whereas RAD21L cohesin complexes dissociate from these regions, REC8 complexes seem to either remain associated with one sister chromatid after dissolution of cohesion or to re-associate with the one sister chromatid. In either case, this REC8 association appears to happen on one sister chromatid only and thus cooperatively. It is likely that reduced acetylation of SMC3 causes increased removal of cohesin complexes by the prophase pathway. In this pathway, which has been described recently for meiocytes (reviewed in Challa et al (2019)), removal of cohesin happens independently of kleisin cleavage involving phosphorylation of REC8. Removal of both, REC8 and RAD21L, was suggested to happen in a prophase-like pathway (Brieno-Enriquez et al, 2016; Wolf et al, 2018). For unknown reasons, ESCO2 impairment affects the removal and thus likely such a prophase pathway most prominently for a RAD21L-based cohesin within the sex body environment. Still, it also affects REC8-based complexes, which, however, have the ability to remain associated with a sister chromatid.

Does the sex chromosome chromatin environment, which appears to form a specific, distinct phase in the nucleus, inhibit synapsis of associated autosomes and thereby affects cohesion or does it inhibit cohesion and thereby affects synapsis? Given that the only known role of ESCO2 is to support cohesion and that cohesion is impaired in the sex body, the most straightforward explanation is that ESCO2-mediated cohesion supports synapsis. As noted above, this is also in agreement with the synapsis defects seen in several cohesin deficiency models. Thus, we prefer a "cohesion-first" model for support of synapsis formation.

$Esco2^{fl/\Delta}$ Vasa-cre male mice are infertile because of incomplete maturation of their spermatids. The reasons for this developmental arrest require a separate investigation. Interestingly, $Esco2^{fl/\Delta}$ Smc1β-icre

male mice produce some sperm and are subfertile. Analysis of excision indicated that no complete excision could be achieved, which suggests that the cells that did not undergo excision were those that produced sperm. In humans, homozygous mutations in $Esco2$ cause a rare developmental disorder called Roberts syndrome (Schule et al, 2005; Vega et al, 2005), but to our knowledge, germ cell development of patients has never been investigated. Given that our results point to a dosage effect of ESCO2 in spermatogenesis, it would be interesting to address whether subfertility in men is associated with polymorphism at the $Esco2$ locus.

Our study demonstrates for the first time in mammals that a cohesion maintenance factor plays an active role during meiotic prophase. This might also be of particular importance for female meiosis, where cohesin maintains chiasmata through sister chromatid cohesion in ageing oocytes for up to several months in mice (Revenkova et al, 2004; Hodges et al, 2005) and possibly up to several decades in women. Although previous studies suggested that expression of meiotic cohesin during dictyate arrest is not necessary (Revenkova et al, 2010) and that no or very little reloading occurs in growing oocytes (Tachibana-Konwalski et al, 2010), continuous SMC3 acetylation may support stability of chromosome-associated associated cohesin and, thus, of cohesion over long periods.

# Materials and Methods

### Mice

$Esco2^{fl/fl}$ mice and $Vasa$-cre mice were described previously (Gallardo et al, 2007; Whelan et al, 2011), the $Smc1β$-iCre strain was newly generated and is based on the $Smc1b$-GFP strain described earlier (Adelfalk et al, 2009), which expresses GFP only in spermatocytes and oocytes. All strains are in the C57BL/6 background. The $Smc1β$-iCre strain was tested by breeding with a $Rosa26$-YFP strain, which carries a floxed STOP codon such that YFP is expressed only after Cre-mediated removal of the STOP codon. FACS analysis showed YFP+ cells only in the testis, not in spleen, kidney, and other control organs; testis section staining showed YFP only in spermatocytes. $Esco2^{fl/fl}$ mice were bred with $Vasa$-cre mice to obtain $Esco2^{fl/+}$ Vasa-cre mice. $Esco2^{fl/+}$ Vasa-cre mice were then backcrossed with $Esco2^{fl/fl}$ mice to obtain $Esco2^{fl/\Delta}$ Vasa-cre mice, whose spermatocytes would be $Esco2^{\Delta/\Delta}$. For genotyping, DNA was extracted from tails using standard procedures. Genotyping was performed by PCR as described in Whelan et al (2011) using an initial denaturation of 7 min at 95°C followed by 35 cycles of denaturation (95°C for 30 s), annealing (57°C for 45 s), and elongation (72°C for 45 s) and a final elongation of 15 min at 68°C and using the following primers: primer 1: 5′-GAC TGG TTT AAT CCT AGG ATA ACT TCG-3′; primer 2: 5′-CTA CCA GTC TTG AGT TCA TGA TGA G-3′; primer 3: 5′-TGT GCA CAT ACT TAT TGA CAG GTG G-3′. Depending on the genotype, the PCR reaction yielded the following products: $Esco2^{fl}$: primers 2 and 3 (product: 347 bp); $Esco2^{\Delta}$: primers 1 and 3 (product: 170 bp); $Esco2^{+}$: primers 2 and 3 (product: 231 bp). Animals were bred and maintained under pathogen-free conditions at the Experimental Center of the Medizinisch-Theoretisches Zentrum of the Medical Faculty at the Technische Universität Dresden according to approved animal welfare guidelines, permission number 24-9168.24-1/2010-25,

24-5131/354/55 granted for the experiments described in this communication by the animal welfare commission of the State of Saxony.

## Analysis of CRE-mediated recombination in FACS-sorted cells

Single-cell suspensions from testes were prepared as described below (see Nuclear spreads and IF). In parallel, single-cell suspensions from the spleens of the same mice were prepared as follows: the spleen of each mouse was collected, cut into small pieces, crushed through a 100-$\mu$m mesh and the filtrate was collected in a tube containing 20 ml of cold PBS. The cells were pelleted by centrifugation for 5 min at 500$g$ at 4°C and red blood cells were lysed by resuspending the cell pellet in 2 ml of ammonium-chloride-potassium (ACK) lysis buffer and incubating for 1 min at 22°C. Lysis was stopped by diluting the ACK lysis buffer with PBS to a final volume of 20 ml, and cells were pelleted by centrifugation as described above. The cells were washed once with 20 ml of PBS, resuspended in 1.2 ml of PBS and kept on ice.

Testis and spleen cells were pelleted by centrifugation, resuspended in 4% PFA and fixed on ice for 5 min. The cells were washed twice with cold PBS and resuspended in 450 $\mu$l of staining solution (0.05 mg/ml propidium iodide [PI], 0.1% trisodium citrate, 0.005% Triton X-100, 0.2 mg/ml RNase A). The cells were stained on ice for 30 min, pelleted by centrifugation, and resuspended in 500 $\mu$l of PBS supplemented with 0.5% BSA and 1 mM EDTA. Cell clumps were removed by filtering through a 40-$\mu$m mesh, and cells were analyzed by FACS and sorted according to their genomic DNA content, which is directly proportional to PI fluorescence intensity. Cells were pelleted by centrifugation and stored at −20°C. The purity of sorted cell populations was assessed by re-analysis by FACS and by immunostaining of samples collected on glass slides using a standard cytospin method. Cell pellets were thawed, resuspended in 500 $\mu$l of lysis buffer (0.2% SDS, 0.1 mg/ml proteinase K, 200 mM NaCl, 5 mM EDTA, and 100 mM Tris, pH 7.5) and incubated for 2 h at 45°C. DNA was extracted with phenol–chloroform followed by chloroform and ethanol/ammonium acetate precipitation in the presence of 10 $\mu$g/ml glycogen as a carrier using standard procedures. DNA was quantified using the NanoDrop and analyzed by PCR as described above.

## Cryosectioning of testes and IF

Immunostaining of testis sections was performed as described recently (Winters et al, 2014) with slight modifications. Briefly, whole testes were fixed for 40 min in fixation solution (4% formaldehyde, 0.1% Triton X-100, and 100 mM sodium phosphate, pH 7.4), washed three times with PBS, incubated overnight at 4°C in 30% sucrose containing 0.02% sodium azide, immersed in O.C.T Compound (Tissue-Tek 4583) in specimen molds (Tissue-Tek 4566 Cryomold 15 × 15 × 5 mm), quickly frozen on dry ice and stored at −80°C. 7-$\mu$m sections were cut using a Leica CM1900 cryostat microtome and placed onto microscope slides (StarFrost K078; 76 × 26 mm). Sections were allowed to dry for 20 min and then immersed in ice-cold methanol for 10 min, then in ice-cold acetone for 1 min, and dried for 10 min. The slides were washed with PBS and then with PBST (PBS containing 0.1% Tween-20) and were subsequently blocked with PBST containing 2% BSA (PBST-2% BSA) for 24 h. The slides

were incubated overnight at 22°C with an anti-SYCP3 antibody (mouse monoclonal, hybridoma cell line supernatant) with or without anti-cIPARP antibody (1:100; No. 9544S; Cell Signaling Technology Inc.). The slides were washed three times with PBST-2% BSA and incubated with a Cy3 goat antimouse IgG (1:300; 405309; BioLegend) for 2 h at 22°C. The slides were washed three times with PBST-2% BSA and mounted using VectaShield mounting medium (H-1000; Vecta Laboratories) containing 1 $\mu$g/ml DAPI and 24 × 50 mm coverslips (Engelbrecht, K12450, thickness of 0.13–0.17 mm).

## Nuclear spreads and IF

Immunostaining of nuclear spreads was performed as described recently (Winters et al, 2014) with slight modifications. Briefly, the tunica albuginea was removed from the testes, and testis tubules were incubated in 500 $\mu$l of 1 mg/ml collagenase for 10 min at 32°C. A single-tubule suspension was obtained by pipetting and then centrifuged for 5 min at 600$g$ at 22°C. The pellet was resuspended in 0.05% trypsin and incubated for 5 min at 32°C with agitation on a rocking platform. Trypsin activity was neutralized by adding 200 $\mu$l of DMEM containing 10% FCS. The single-cell suspension was filtrated through a 40-$\mu$m strainer by centrifugation at 1,200$g$ for 10 s and then centrifuged at 600$g$ for 5 min at 22°C. The pellet was resuspended in 500 $\mu$l of PBS. 1.5 $\mu$l of single-cell suspension was added to each well of a 10-well slide (ER-308B-CE24; Thermo Fisher Scientific, 10 well, 6.7 mm) containing 7 $\mu$l of 0.25% NP-40; cells were lysed for 2 min and fixed by adding 25 $\mu$l of fixation solution (1% PFA and 10 mM sodium borate, pH 9.2). The slides were incubated in a wet chamber for 1 h, dried for 30 min to 1 h, washed two times with 0.5% Photo-Flo (146 4510; KODAK), washed three times with H$_2$O, dried, and stored at −20°C until they were used. Nuclear spreads for SIM (see below) were prepared using the same procedure, except that the spreads were prepared directly on coverslips instead of using 10-well slides.

For IF, primary antibodies used were as follows: mouse anti-SYCP3 (undiluted, hybridoma supernatant), rabbit anti-SYCP3 (1:500, NB300-230; Novus Biologicals), rabbit anti-acSMC3 (1:10, MBL PD040; raised against the acetylated synthetic peptide CIGA(acK)(acK)DQYFL, which corresponds to the conserved sequence including K105/K106 of SMC3), rabbit anti-SMC3 (1:100, A300-060A; Bethyl Laboratories), rabbit anti-sororin (1:25, kindly provided by JA Suja and JL Barbero), rabbit antimouse ESCO2 (1:25, [Whelan et al, 2011]), guinea pig antimouse ESCO2 (1/5, [Whelan et al, 2011]), rabbit antihuman ESCO2 (1:1,000, A301-689A; Bethyl Laboratories), guinea pig anti-REC8 (1:50 or 1:100; kindly provided by Dr Christer Höög), rabbit anti-REC8 (1:100; kindly provided by Dr Melina Schuh); mouse anti-γH2AX (1:700, 05-636; Millipore), mouse anti-γH2AX biotin conjugate (1:2,000, 16-193; Millipore), rabbit anti-H1t (generated in the lab of Dr Peter Moens and kindly provided by Dr Edyta Marcon), rabbit anti-SYCP1 (1:100, ab15090; Abcam), rabbit anti-DMC1 (1:50, sc-22768; Santa Cruz Biotechnology), human anti-centromere (1:5, 15-235-0001; Antibodies Incorporated), rabbit anti-RAP1 (1:50, IMG-289; Imgenex), rabbit anti-RAD21 (1:100, ab992; Abcam), two independent rabbit anti-RAD21L (each 1:100, kindly provided by Dr A Pendas and Dr T Hirano), rabbit anti-MAU2 (1:120, ab46906; Abcam), rabbit anti-MVH (1:1,000, ab13840; Abcam), rabbit anti-RNA Pol II (1:2,000, ab5408; Abcam), and rabbit anti-RAD51

(1:100, GTX100469-100; Genetex). We like to point out that the anti-ESCO2 antibody Bethyl Laboratories 301-689A used in Evans et al (2012) and some other publications does not yield a specific signal because it recognizes an unknown antigen present in ESCO2-deficient MEFs (Fig S4). Secondary antibodies were used as follows: Cy3 goat antimouse IgG (1:300, 405309; BioLegend), Alexa Fluor 488 goat antirabbit IgG (1:500, A11034; Invitrogen), FITC goat antimouse (1:300, 101002; Southern Biotechnology), Alexa Fluor 488 goat anti–guinea pig (1:500, A11073; Invitrogen), Alexa Fluor 555 goat antirabbit (1:500, A21428; Invitrogen), and Alexa Fluor 568 goat antihuman IgG (1:300, A21090; Invitrogen).

## Microscopy and imaging

Except where stated otherwise, conventional fluorescence microscopy was performed using an Olympus (JX70) or a Zeiss Axiophot microscope with a 40× or 100× objective (with oil of refractive index 1.518 [Immersol 518 F; Carl Zeiss]) to obtain a magnification of 400× or 1,000×, respectively. Images were acquired using the AxioVision Rel. 4.6.3.0 software and, when necessary, were processed and analyzed using Fiji, which was also used to quantify signal intensities by selecting the DAPI-positive region. Statistics was performed using the unpaired $t$ test. Z-stacks were generated on the Zeiss LSM880 Airyscan microscope using in average 0.2-$\mu$m distance between stacks, generating 14–16 stacks per sample.

SIM was performed at the microscopy facility of the Max Planck Institute of Molecular Cell Biology and Genetics (Dresden, Germany) using the Deltavision OMX v3 BLAZE (Applied Precision Inc. [API]) with an Olympus PlanApo N 60× objective (1.42 Oil UIS2 inf/0.17/FN26.5 WD 0.15 mm). The API software and OMX software were used for driving the microscope and super-resolution reconstruction. For OMX SI reconstruction, a Linux Box was used (centos 4, 10.1.145.31 personalDV linux centos5 box 10.1.145.32). Before performing SIM, nuclei were visualized and imaged by conventional microscopy (also available on the OMX system) and selected according to their meiotic prophase substage. For both conventional and structured illumination images produced with the OMX system, Z projections consisting of the sum of slices were generated using Fiji.

## Immunoprecipitation from testis nuclear extracts

For immunoprecipitation experiments, nuclear protein extracts from testes were prepared essentially as described in Jessberger et al (1993), Winters et al (2014). Protein extracts and immunoprecipitation reactions were prepared in parallel from control (*Esco2*$^{fl/+}$) and *Esco2*$^{fl/\Delta}$ *Vasa-cre* mice. Briefly, the tunica albuginea was removed from the testes and single-cell suspensions were created using Dounce homogenization (loose pestle) in buffer B (5 mM KCl, 2 mM DTT, 40 mM Tris [pH 7.5], 2 mM EDTA, and protease inhibitors). Nuclear membranes were lysed using Dounce homogenization (tight pestle). Nuclear suspensions were centrifuged at 1,180$g$ for 3 min, nuclear pellets were resuspended in buffer C (5 mM KCl, 0.1 mM DTT, 15 mM Tris [pH 7.5], 0.5 mM EDTA, and protease inhibitors), and nuclear proteins were extracted by adding ammonium sulfate (pH 7.4) to a concentration of 250 mM and incubating on ice for 30 min. The samples were centrifuged at 100,000$g$ for 30 min at 4°C. Supernatants were collected, and protein contents were measured using the NanoDrop. For each immunoprecipitation reaction, 600 $\mu$g of

nuclear extract was mixed with the same volume of anti-SMC1$\beta$ IgG (mouse hybridoma supernatant) or anti-MYC IgG (also mouse hybridoma supernatant, used as control IgG) and incubated overnight at 4°C on a rotating wheel. Protein G Dynabeads (100-03D; Invitrogen) were then added, and immunoprecipitations were performed according to the manufacturer's instructions. Immunoprecipitates were loaded onto SDS–PAGE gels together with their respective inputs (5% of the amount of protein used for IP, kept at 4°C during the entire procedure) and analyzed by immunoblotting.

## Extraction of total proteins from FACS-sorted testis cells and real-time PCR

Testis cells were prepared, stained with Hoechst 33342, and sorted by FACS as described in Bastos et al (2005) with slight modifications. Briefly, the tunica albuginea was removed and tubules were incubated for 20 min at 32°C in 1 ml of digestion buffer (0.4 mg/ml collagenase in HBSS supplemented with 20 mM Hepes [pH 7.2], 1.2 mM MgSO$_4$ 7H$_2$O, 1.3 mM CaCl$_2$ 2H$_2$O, 6.6 mM sodium pyruvate, and 0.05% lactate). Interstitial cells were partly eliminated by carefully removing the supernatant and tubules were dissociated by pipetting up and down in 1 ml of fresh digestion buffer and were incubated for 20 min at 32°C. A single-cell suspension was prepared by filtering through a 40-$\mu$m strainer by centrifugation at 800$g$ for a few seconds and then centrifuged at 600$g$ for 5 min at 22°C. The cell pellet was resuspended in 20 ml of incubation buffer (HBSS supplemented with 20 mM Hepes [pH 7.2], 1.2 mM MgSO$_4$ 7H$_2$O, 1.3 mM CaCl$_2$ 2H$_2$O, 6.6 mM sodium pyruvate, 0.05% lactate, 1 mM glutamine, and 1% fetal calf serum) and kept on ice for 20 min. Hoechst 33342 was added at a concentration of 5 $\mu$g/ml, and the cells were stained at 32°C for 1 h. The cells were then centrifuged for 5 min at 600$g$ at 22°C, resuspended in 4 ml of incubation buffer, and analyzed/sorted by FACS as described in Bastos et al (2005). Sorted cells were harvested in cold PBS and reanalyzed to confirm purity. The cells were collected by centrifugation for 15 min at 600$g$ at 4°C, supernatants were discarded, and cell pellets were snap-frozen in liquid nitrogen and stored at −80°C.

Total protein extracts from sorted and unsorted populations were prepared as follows: the cell pellets were thawed on ice and resuspended in 25 $\mu$l of radioimmunoprecipitation assay (RIPA) buffer (150 mM NaCl, 0.1% SDS, 1% NP-40, 0.5% sodium deoxycholate, and 50 mM Tris, pH 7.5) supplemented with 1 mM DTT, 1 mM PMSF, and protease inhibitors. The cells were lysed for 20 min on ice, and the cell lysates were sonicated at 4°C using a Branson Sonifier 450 sonicator (6 pulses of 20 s with 1 min intervals). Cell extracts were centrifuged at 16.000$g$ for 15 min at 4°C, 20 $\mu$l of the supernatants (total protein extracts) were collected, mixed with 5 $\mu$l of 5× Laemmli buffer, run on a 7.5% SDS–PAGE gel, and analyzed by immunoblotting.

For real-time RT-PCR, testis cells were prepared, stained with Hoechst 33342, and sorted by FACS as described above. Cell pellets of sorted cell populations were lysed in 750 $\mu$l Trizol reagent (15596026; Invitrogen) and stored at −80°C until RNA was extracted. RNA isolation was performed according to the manufacturer's protocol with adjusted volumes and one additional wash with cold 75% ethanol. RNA concentrations were measured using the NanoDrop. RNA was kept at −80°C or used directly for cDNA synthesis. Reverse transcription was performed using M-MLV

reverse transcriptase (RNase H minus, point mutant Cat. no. M3683; Promega) and Oligo(dT)$_{15}$ primers (Cat. no. C1101; Promega) according to the manufacturer's protocol. cDNA was either used directly for real-time PCR or stored at −20°C until use. 10 ng cDNA per reaction were used. Triplicates of each sample were performed. Real-time PCR was performed using Rotor Gene SYBR Green PCR Kit (204076; QIAGEN) according to the manufacturer's protocol, and 2 pmol primers were added (primer sequences: TATA box binding protein as housekeeping gene [Tbp]: fwd GCAGTGCCCAGCATCACTAT, rev TGGAAGGCTGTTGTTCTGGT and Esco1 fwd CAGCACCAGATCAGAATTTCAG, rev GGAGCTGAACCTGGAAATGT). Real-time PCR was run on qTower2.0 (Analytik Jena) using qPCRsoft2.1 software for analysis.

## Immunoblotting

Proteins were transferred onto a nitrocellulose membrane and blocked in 5% milk in PBST for 30 min at 22°C. Membranes were reacted for 2–4 h at 22°C or overnight at 4°C with primary antibodies diluted in PBST. Primary antibodies used were as follows: rabbit anti-acSMC3 (1:1,000, MBL), rabbit anti-SMC3 (1:1,000, A300-060A; Bethyl Laboratories), mouse anti-SMC1$\beta$ (1:2, hybridoma supernatant), guinea pig antimouse ESCO2 (1:500, [Whelan et al, 2011]), and mouse anti-GAPDH (1:200, sc-32233; Santa Cruz). Membranes were washed three times in PBST, and HRP-conjugated secondary antibodies were added for 1 h at 22°C in PBST. Secondary antibodies were all diluted 1:5,000 and were used as follows: antirabbit IgG HRP (18-8816-31; eBioscience), antimouse IgG HRP (115-035-003; Dianova), and anti–guinea pig IgG HRP (106-035-003; Dianova). A prestained protein ladder (#26619; Thermo Fisher Scientific) was loaded onto each gel to determine the molecular weight of the detected bands. Blots were washed three times in PBST and developed using chemiluminescent HRP substrate (WBKLS0100; Millipore) and imaged on a Kodak ImageStation 2000MM.

# Supplementary Information

# Acknowledgements

We are grateful to Dr Attila Toth for helpful discussions. We thank Dr Glen Pearce for help with FACS sorting. We thank the following colleagues for sharing antibodies: Dr JA Suja and Dr JL Barbero (anti-sororin), Dr A Pendas (anti-RAD21L), anti-H1t (Dr Edyta Marcon/the late Dr Peter Moens), Dr Christer Höög (anti-REC8), and Dr Melina Schuh (anti-REC8). This work was funded by a grant from the European Union (H2020, GermAge) and from the Deutsche Forschungsgemeinschaft (DFG, JE 150/22-1) to R Jessberger.

## Author Contributions

F McNicoll: conceptualization, resources, data curation, formal analysis, validation, investigation, visualization, methodology, and writing—original draft, review, and editing.

A Kühnel: conceptualization, resources, data curation, formal analysis, validation, investigation, visualization, methodology, and writing—original draft, review, and editing.
U Biswas: resources, data curation, validation, investigation, visualization, methodology, and writing—review and editing.
K Hempel: data curation, investigation, and methodology.
G Whelan: resources, data curation, investigation, methodology, and writing—review and editing.
G Eichele: conceptualization, resources, supervision, funding acquisition, and writing—original draft, review, and editing.
R Jessberger: conceptualization, resources, data curation, formal analysis, supervision, funding acquisition, validation, investigation, visualization, methodology, project administration, and writing—original draft, review, and editing.

## Conflict of Interest Statement

The authors declare that they have no conflict of interest.

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
