## [Reviewer comments · Life Science Alliance]

Meiotic Sex Chromosome Cohesion and Autosomal Synapsis are Supported by Esco2

François McNicoll, Anne Kühnel, Uddipta Biswas, Kai Hempel, Gabriela Whelan, Gregor Eichele and Rolf Jessberger

DOI: 10.26508/lsa.202000564

Corresponding author(s): Prof. Rolf Jessberger (Dresden University of Technology)

Review timeline:

Submission Date:	2019-09-28
Editorial Decision:	2019-10-31
Revision Received:	2020-01-23
Editorial Decision:	2020-01-24
Revision Received:	2020-01-29
Accepted:	2020-01-31

Scientific Editor: Andrea Leibfried

Transaction Report:

No Peer Review Process File is available with this article, as the authors have chosen not to make the review process public in this case.

October 31, 2019

Re: Life Science Alliance manuscript #LSA-2019-00564-T

Prof. Rolf Jessberger
Dresden University of Technology
Inst. of Physiological Chemistry
Fiedlerstr. 42
Dresden 1307
Germany

Dear Dr. Jessberger,

Thank you for submitting your manuscript entitled "Meiotic Sex Chromosome Cohesion and Autosomal Synapsis are Supported by Esco2" to Life Science Alliance. The manuscript was assessed by expert reviewers, whose comments are appended to this letter.

As you will see, the reviewers appreciate your work and provide constructive input on how to further strengthen it. We would thus like to invite you to submit a revised version to us, addressing the individual points raised. This will require mainly text changes / restructuring and considering alternative hypotheses, but also a few experiments (please repeat the Rec8 staining and provide antibody validations as requested).

Thank you for this interesting contribution to Life Science Alliance. We are looking forward to receiving your revised manuscript.

Sincerely,

Andrea Leibfried, PhD

Executive Editor
Life Science Alliance
Meyerohofstr. 1
69117 Heidelberg, Germany
t +49 6221 8891 502
e a.leibfried@life-science-alliance.org
www.life-science-alliance.org

B. MANUSCRIPT ORGANIZATION AND FORMATTING:

2nd Editorial Decision

24 January 2020

January 24, 2020

RE: Life Science Alliance Manuscript #LSA-2019-00564-TR

Prof. Rolf Jessberger
Dresden University of Technology
Inst. of Physiological Chemistry
Fiedlerstr. 42
Dresden 1307
Germany

Dear Dr. Jessberger,

Thank you for submitting your revised manuscript entitled "Meiotic Sex Chromosome Cohesion and Autosomal Synapsis are Supported by Esco2". I assessed your revised manuscript and your response to the reviewer concerns, and I think that these were adequately addressed. We would thus be happy to publish your paper in Life Science Alliance pending final revisions necessary to meet our formatting guidelines:

- Please make sure that the author order in your manuscript and in our submission system match
- Please add the author contributions for each author in our submission system
- Please add a callout to figure 3A in your manuscript text
- Please add scale bars to Fig S4, S9A, S16
- Please provide the source data for Fig. S6A

A. FINAL FILES:

-- Summary blurb (enter in submission system): A short text summarizing in a single sentence the study (max. 200 characters including spaces). This text is used in conjunction with the titles of papers, hence should be informative and complementary to the title. It should describe the context and significance of the findings for a general

readership; it should be written in the present tense and refer to the work in the third person. Author names should not be mentioned.

B. MANUSCRIPT ORGANIZATION AND FORMATTING:

Sincerely,

Andrea Leibfried, PhD
Executive Editor
Life Science Alliance
Meyershofstr. 1
69117 Heidelberg, Germany
t +49 6221 8891 502
e a.leibfried@life-science-alliance.org
www.life-science-alliance.org

3rd Editorial Decision

31 January 2020

January 31, 2020

RE: Life Science Alliance Manuscript #LSA-2019-00564-TRR

Prof. Rolf Jessberger
Dresden University of Technology
Inst. of Physiological Chemistry
Fiedlerstr. 42
Dresden 1307
Germany

Dear Dr. Jessberger,

Thank you for submitting your Research Article entitled "Meiotic Sex Chromosome Cohesion and Autosomal Synapsis are Supported by Esco2". It is a pleasure to let you know that your manuscript is now accepted for publication in Life Science Alliance. Congratulations on this interesting work.

DISTRIBUTION OF MATERIALS:

Again, congratulations on a very nice paper. I hope you found the review process to be constructive and are pleased with how the manuscript was handled editorially. We look forward to future exciting submissions from your lab.

Sincerely,

Andrea Leibfried, PhD

Executive Editor
Life Science Alliance
Meyerhofstr. 1
69117 Heidelberg, Germany
t +49 6221 8891 502
e a.leibfried@life-science-alliance.org
www.life-science-alliance.org